# Development and Initial Validation of the Korean Effort and Reward Scale (ERS-K) for Use in Sport Contexts

**DOI:** 10.3390/ijerph182413396

**Published:** 2021-12-20

**Authors:** InKyoung Park, YoungHo Kim

**Affiliations:** Department of Sport Science, Seoul National University of Science and Technology, Seoul 01811, Korea; pik77812@nate.com

**Keywords:** development, validation, effort, reward, scale, university athletes

## Abstract

Background: Applying Siegrist’s (1996) effort-reward imbalance model to athletes, the current study aimed to develop a measure that can evaluate athletes’ effort and reward, and verify its reliability and validity. Methods: The survey was conducted on 530 athletes from universities in Seoul, South Korea. Among the collected data, 276 datasets were used for exploratory factor analysis, 200 for confirmatory factor analysis, and 30 for test-retest reliability analysis; data from surveys that were incomplete or incorrectly answered were excluded. The initial questionnaire was completed based on prior research, expert meetings, and evaluation by the evaluation group. The collected data were subjected to question analysis, exploratory factor analysis, confirmatory factor analysis, reliability analysis, and criterion-related validity analysis. Result: Four factors of the construct of effort were extracted: training strengthening efforts, interpersonal efforts, nutrition management efforts, and cognitive psychology strengthening efforts. Additionally, three factors of the construct of reward were extracted: future stability, social support, and positive growth. Thus. the effort measurement tool was finalized with 14 questions from four factors, and the reward measurement tool was finalized with 14 questions from three factors, with all items rated on a 5-point Likert scale. Conclusions: Siegrist’s efforts to measure job stress and athletes perceived efforts differed somewhat, but were found to be consistent with those reported for Australian occupational jockeys. In addition, athletes’ perceived rewards had similar results to those for Siegrist and Kathleen et al. studies. Based on this study, subsequent studies can more effectively determine whether the effort-reward imbalance model is applicable to athletes.

## 1. Introduction

In general, adults who have completed their studies take up a job to ensure their survival. Efforts for survival are essential for humans, who are social beings, and humans make considerable quantitative and qualitative efforts to not fall behind in the process of the rapid development of society. They also expect to receive the right reward for their efforts. However, humans may be exposed to many stressful situations as the rewards may not always be equal to the efforts made by them. 

The effort-reward imbalance (ERI) model has been developed to explain psychological and physical health-related job stress in adults [1]. The ERI model implies that it is important to balance efforts and rewards for individual health. For example, if the balance between efforts and rewards is incongruent and ERI is perceived as high, it negatively affects psychological factors, such as well-being, self-esteem, and self-efficacy in the individual, causing stress and lowering the tendency for continuous engagement in a given task [2]. In addition, Siegrist [3] argued that studies applying the ERI model provide significant insight into the socio-psychological factors affecting individual health. 

According to previous studies, ERI generates negative affect and stress response in an individual when they receive a small reward despite significant effort, whereas appropriate social rewards play an important role in well-being, health, and performance continuation by promoting positive affective responses [4,5,6]. Thus far, numerous studies have applied the concept of ERI to various occupational groups such as doctors [7], nurses [8,9], office workers [10], teachers [11], hotel workers [12], media workers [13], police officers [14], and university students [15]. ERI has been found to have a significant effect on occupational performance [16] and competence [17], physical and psychological diseases [18], alcohol-related problems [19], and workplace harassment [20]. 

From this point of view, some studies have pointed out that financial compensation, such as pension or annual salary, is of great significance, even in the field of sports, and can have a significant effect on athletic performance [21,22]. Duncan and Craig [23] argued that rewards influence the expectations of athletes and act as a motive for enduring lonely and difficult training. Choi [24] stated that although some athletes are excessively obsessed with financial rewards and neglect sportsmanship, causing problems such as game manipulation and sports gambling, external rewards, whether positive or negative, affect the performance of athletes. While several studies have reported a significant effect of rewards on the performance of athletes, there has been insufficient research on how the effort-reward balance and imbalance perceived by athletes affects their performance and related psychological attributes. 

According to Siegrist [1], most people endure ERI when: there is no other alternative to avoid the imbalance in the current situation; they have to accept the imbalance for strategic reasons, such as expectations for future gains; or when they have a motive, such as being overly immersed in work. In this respect, one of those who frequently endures ERI over a long period of time is an athlete. Athletes, for example, accept ERI due to their expectations for good performance at major competitions (such as the Olympics and world championships) and the associated benefits. 

However, several studies have reported that athletes experience exhaustion due to decreased internal and external rewards, such as decreased desire for achievement, depersonalization, and devaluation by coaches and teammates [25,26]. Athletes invest a lot of effort to obtain the best results and rewards while accepting ERI, and continued investment of efforts in this state of ERI can cause mental and physical exhaustion. In other words, rewards in the field of sports are significantly related to good performance and prolonged athletic careers, which suggests that ERI can act as an important factor influencing the participation, immersion, exhaustion, and abandonment of athletes. Therefore, identifying the level of ERI perceived by athletes and the effect of this perceived ERI on the performance of athletes will greatly contribute, not only to improving the performance of athletes in the sports field, but also in preparing an effective psychological support program for them. 

Effort-reward balance is considered important to induce positive changes in the performance of athletes and their psychological attributes in the sports field. However, an objective tool to measure perceived efforts and rewards in athletes has not been developed. Therefore, this study aimed to develop an effort and reward scale for athletes, and to evaluate its reliability and validity.

## 2. Materials and Methods

### 2.1. Participants

To develop an effort and reward scale for athletes, a total of 530 athletes registered at universities in Seoul, Gyeonggi-do, Gangwon-do, and Jeolla-do for participation. Of the data collected from the 530 athletes, 24 datasets had the same responses throughout, had no response for more than three items, or did not include information on personal characteristics; hence, these datasets were not used in the study. Data from 276 athletes were used in the process of extracting the factors of the constructs of effort and reward (male: 192, female: 84, average age: 19.89, average athlete career: 8.51), and data from 200 athletes were used for model verification (male: 142, female: 58, average age: 20.03, average athlete career: 8.69). The remaining 30 datasets were used for the test-retest reliability analysis (male: 30, average age: 20.23, average athlete career: 9.9).

In addition, expert opinions were collected at a meeting of eight experts including two professors and six doctors in sports and exercise psychology for creating the initial items of the questionnaire and deciding the factor names after searching for the contents.

### 2.2. Development Methods and Procedures

Figure 1 shows the procedure for creating a scale of the efforts and rewards perceived by athletes.

#### 2.2.1. Creation of Initial Items

In this study, a scale was developed to quantify the efforts and rewards of athletes using the following process: First, the concept and structure of the efforts and rewards of athletes were identified, and the factors were specified. The content was created based on an empirical search of the efforts and rewards presented in a previous study [27]. In a study by Park et al. [27], which confirmed the concept of athlete efforts and rewards, athletes’ efforts were explored in terms of skill enhancement, self-management, cognitive psychological reinforcement, and interaction, and athletes’ rewards were extracted in terms of benefits, social support, positive outcomes, and future stability. Based on the content, and through meetings with the group of experts, as many initial items as possible were developed to interrogate the athletes regarding the attributes of efforts and rewards. All the items were then reevaluated by a process of correcting, supplementing, and deleting ambiguous or semantically redundant questions (efforts: 49 items, rewards: 38 items).

#### 2.2.2. Validity Evaluation

The validity of the effort and reward scale for athletes was evaluated through exploratory factor analysis (EFA), confirmatory factor analysis (CFA), comparison of mean differences between groups, and correlation analysis. To determine the logical relationship between the finally determined factors and other tools through EFA and CFA, a questionnaire for the assessment of self-management practices of athletes developed by Kim [28] was used. The questionnaire for the assessment of self-management practices of athletes comprised 26 items on the six factors of mental readiness, training management, physical conditioning, interpersonal relationships, personal routine, and daily living management. The answers were assessed on a 5-point Likert scale (1 = strongly disagree to 5 = strongly agree). A higher score indicated better self-management practices. The reliability and validity of this scale have been determined in various research studies; the Cronbach’s α for the overall items in this study was 0.88, with a reliability of 0.64–0.88 for the factors. The fit of the measurement model was also good [x^2^(212) = 369.99, *p* < 0.001, RMSEA: 0.06, RMR: 0.04, TLI: 0.90, CFI: 0.92].

#### 2.2.3. Reliability Evaluation

To determine reliability, the internal consistency of items using Cronbach’s α and test-retest reliability were measured. For test-retest reliability, data were collected from one college football team using the same administration method with a gap of two weeks between the test and retest. The data collected for the test-retest were analyzed for mean difference and correlation, and there was little difference in the means of the scores collected at the two time points; if there was a significant correlation, it was judged that there was temporal stability. 

### 2.3. Research Procedure

The survey with the athletes was conducted in comfortable resting spaces at the residence hall or training ground of the team. Prior to responding to the questionnaire, the researcher explained the purpose and content of the study, as well as the content of the questionnaire. The participants were asked to sign consent forms if they wished to participate in the study. In addition, it was clearly stated that the personal information obtained from the questionnaire would not be used for purposes other than research, and that the research participants could withdraw from participation at any time they wanted. It took approximately 15 minutes for them to complete the survey, and the completed questionnaires were collected individually. This research process was conducted with the approval of the Institutional Review Board (IRB) of Seoul National University of Science and Technology (IRB approval number: 2020-0003-01).

### 2.4. Data Analysis

The data collected in this study were first used for item analysis (item distribution and basic statistics) using SPSS (version 23.0; IBM Corp., Armonk, NY, USA). Internal structure validity was determined by EFA using Jamovi 1.6.15 [29]. AMOS 23.0 (IBM Corp., Armonk, NY, USA) was used for CFA. Specifically, for EFA, factors were extracted using the maximum likelihood method of common factor analysis based on the parallel analysis method. Oblique rotation, which assumes a correlation between factors, was used for factor rotation. In the evaluation of the adequacy of the factor structure, factor loads, eigenvalues, screen plots, and explanatory quantities were considered. For CFA, the suitability and standardized coefficients of the model were calculated to verify the theoretical validity of the structural model. 

Next, to verify the reliability of the effort and reward scale for athletes, Cronbach’s α was first calculated to determine the internal consistency of the items. Thereafter, 30 participants were subjected to test-retest (paired t-test and correlation analysis) with a gap of two weeks between the test and retest. Finally, the discriminant validity and convergent validity of the finally determined factor structure were examined to determine the differences in the efforts and rewards of players according to gender, competition performance, and athletic career (multivariate analysis of variance, MANOVA); the correlation with self-management was also analyzed. Reliability evaluation, MANOVA, and correlation analysis were performed using the SPSS 23.0 program.

## 3. Results

### 3.1. Construct Validity of Effort and Reward

#### 3.1.1. Item Analysis of Effort and Reward

Item analysis (mean, standard deviation, skewness and kurtosis) was conducted to check the distribution of the athlete’s effort and reward items. The mean of each item ranged from 2.54 to 4.41, and the standard deviation ranged from 0.66 to 1.37. Examining the skewness and kurtosis to determine the degree of distribution for each item confirmed that most of the values were close to “0” and did not exceed ±2, indicating that the distribution of the items was within the normal distribution range. Therefore, all the items were used for EFA.

#### 3.1.2. EFA of Effort and Reward

To determine the effort and reward structure of the athletes, EFA was performed using maximum likelihood and oblique rotation of the common factor analysis based on parallel analysis. The details for the EFA of effort and EFA of reward are individually presented below.

##### EFA of Effort

For the initial effort, the suitability index of Kaiser–Meyer–Olkin (KMO) was 0.90, which was close to one, and the correlation between the samples was judged to be good. The result of Bartlett’s test of sphericity was 6192.41 (*p* < 0.001), indicating that the data were suitable for factor analysis. However, no convergence was observed in the factor rotation. Therefore, items with an item-total correlation below 0.20, and items that belonged to several factors in the factor analysis were deleted. In addition, gradual deletion was considered while reviewing the construct and meaning of the items. 

As a result, 14 items from four factors were extracted. Each factor load ranged from 0.33 to 0.90, and the total variance of the four factors was 51.3%. The reliability for each of the four factors was 0.66–0.85, and the total reliability was 0.85. The results of the item analysis, EFA, and reliability analysis are presented in Table 1. The KMO for the final extraction factor was 0.843, and the result of Bartlett’s test of sphericity was 1353.32 (*p* < 0.001).

##### EFA of Reward

For the initial effort, the suitability index of KMO was 0.92, which was close to one, and the correlation between the samples was judged to be good. The result of Bartlett’s test of sphericity was 5153.83 (*p* < 0.001), indicating that the data were suitable for factor analysis. However, items whose deletion would have resulted in a higher Cronbach’s α value than the overall reliability, and items with an item-total correlation below 0.20 were deleted. Items belonging to several factors in the factor analysis were also deleted. In addition, gradual deletion was considered while reviewing the construct and meaning of the items. 

As a result, 14 items from three factors were extracted. Each factor load ranged from 0.50 to 0.89, and the total variance of the three factors was 51.7%. The reliability for each of the four factors was 0.76–0.85, and the total reliability was 0.85. The results of the item analysis, EFA, and reliability analysis are presented in Table 2. The KMO for the final extraction factor was 0.90, and the result of Bartlett’s test of sphericity was 1669.0 (*p* < 0.001).

#### 3.1.3. CFA of Effort and Reward

CFA was performed using the AMOS 23.0 program to determine the validity of the structural model for the 14 items of the four effort factors and 14 items of the three reward factors obtained by EFA.

##### CFA of Effort

With respect to the fit of the model for effort, χ^2^ (71) was 164.03 (*p* < 0.001), RMSEA was 0.07, SRMR was 0.06, TLI was 0.91, and CFI was 0.93. CFI and TLI were 0.9 or higher, and the absolute fit index, RMSEA, which considered the simplicity of the model, was 0.08 or less, indicating a good fit [30]. Information on the relationship between factors and items appeared to support the hypothetical structure explored. The regression coefficients for each factor are shown in Table 3.

The validity of the CFA was ensured by evaluating construct validity, convergent validity, and discriminant validity. Construct validity was evaluated using the standard regression weight (SRW) and squared multiple correlations (SMC) between the measured variables. Convergent validity was evaluated by calculating the average variance extracted (AVE) and construct reliability (CR). In general, an SRW of 0.5 or higher and a CR of ±1.96 or higher indicate that the measured variables are well explained. Furthermore, an AVE of 0.5 or higher and a CR of 0.7 or higher indicate acceptable convergent validity [31]. Finally, discriminant validity is determined by the confidence interval (ø ± 2Xstandard error) of the correlation coefficient not including 1.0, and a correlation coefficient of 0.9 or lower between factors [32]. 

The results indicated the AVE as lower than 0.5 for factors 3 (0.49) and 4 (0.46), and SMC lower than 0.4 for effort item 12 (0.40). The SRWs were 0.5 or higher, CR was greater than ±1.96, and the CR value was also 0.7 or higher, indicating that the construct validity and convergent validity were ensured in general. On examining whether the correlation coefficient was 0.9 or lower between factors, and whether the confidence interval of the correlation coefficient included 1.0, for the highest correlation coefficient of 0.63, (ø ± 2 × standard error = 0.63 ± 2 × 0.07), the confidence interval was found to be 0.50–0.76, indicating that 1.0 was not included.

##### CFA of Reward

With respect to the fit of the model for reward, $x^{2}$ (74) was 158.22 (*p* < 0.000), RMSEA was 0.08, SRMR was 0.06, TLI was 0.91, and CFI was 0.93. CFI and TLI were 0.9 or higher, and the absolute fit index, RMSEA, which considered the simplicity of the model, was 0.08 or less, indicating a good fit [30]. Information on the relationship between factors and items appeared to support the hypothetical structure explored. The regression coefficients of each factor for reward are shown in Table 4. 

The validity of the CFA for reward was determined in the same manner as for the effort factors. The results revealed the AVE as lower than 0.5 for factor 2 (0.47), and the SMC as lower than 0.4 for reward item 2 (0.39). The SRWs were 0.5 or higher, CR was greater than ±1.96, and the CR value was also 0.7 or higher, indicating that the construct validity and convergent validity were ensured in general. On examining whether the correlation coefficient was 0.9 or lower between factors, and whether the confidence interval of the correlation coefficient included 1.0, for the highest correlation coefficient of 0.8, (ø ± 2 × standard error = 0.80 ± 2 × 0.07) the confidence interval was found to be 0.69–0.90, indicating that 1.0 was not included [32].

#### 3.1.4. Discriminant Validity and Convergent Validity of Effort and Reward

##### Differences between Groups

To evaluate the discriminant validity of the effort and reward scale, differences according to gender, recent winning status, and athletic career were examined by MANOVA. For the analysis of differences based on recent winning status, the athletes were classified into those with any experience of winning and those without an experience of winning. Differences based on athletic career were analyzed by classifying the athletes into those with eight years of experience or less, and those with nine years of experience or more, based on the mean professional experience of the participants. The basic statistics and results of MANOVA are shown in Table 5. 

Looking at the difference between the factors by group, males showed statistically significantly higher scores in effort factor 1, effort factor 2, and effort factor 4 compared to females. The athletes with nine years of experience or more showed a higher score in effort factor 3 than those with eight years of experience or less. In addition, there were no statistically significant differences between the groups in terms of reward factors. 

##### Correlation with Another Scale

The correlation between the effort and reward scale and the self-management scale was used to evaluate the convergent validity of the former. As shown in Table 6, the highest correlation in the correlation analysis between the factors of effort and reward and the factors of self-management was found between daily living management and interpersonal relationships. The factors of the effort and reward scale mostly correlated with the factors of the self-management scale. 

Table 5 shows the results of two-way repeated-measures ANOVA to examine the effects of the intervention on psychological variables of participants. The results indicated that participants in the experimental group displayed significant increase in pros [*F*(1, 58) = 51.47, *p* < 0.001], self-efficacy [*F*(1, 58) = 49.31, *p* < 0.001], cognitive processes [*F*(1, 58) = 17.17, *p* < 0.01], behavioral processes [*F*(1, 58) = 12.32, *p* < 0.01], and cons [*F*(1, 58) = 3.77, *p* < 0.05] over the intervention. Additionally, participants in the experimental group showed significantly higher scores on most of the psychological variables, except cons, than those in the control group after the intervention [*F*(1, 58) = 79.45 for self-efficacy, *p* < 0.001; *F*(1, 58) = 63.13 for pros, *p* < 0.001; *F*(1, 58) = 29.97 for behavioral processes, *p* < 0.001; and *F*(1, 58) = 17.17 for cognitive processes, *p* < 0.01].

### 3.2. Reliability Evaluation for the Effort and Reward Scale

Next, a test-retest was conducted to evaluate the temporal stability of the effort and reward scale for athletes. For the analysis, a survey was completed twice by 30 athletes from among the participants at intervals of two weeks, and a paired t-test and correlation analysis between the two response scores were performed Table 7. The paired t-test showed no difference in all factors, and significant correlations were found for all factors. There was no mean difference between the test and retest scores, and the correlation coefficient was high, indicating that the effort and reward scale for athletes demonstrates temporal stability.

### 3.3. Factors of the Constructs of Effort and Reward

Upon identification of the constructs of effort and reward for athletes, and development of a measuring tool to evaluate these constructs, the construct of effort was manifested as four factors. The identified factors were named based on expert opinions and a literature review. The first factor was named “training reinforcement efforts”, as it included the items: “I train individually (besides the team training)”, “I do extra-training sessions before and after scheduled team training”, “I try more than other athletes”, “I do additional strength and conditioning training on my own” and “I spend more time for training than other athletes do”.

The second factor was named “interpersonal relationship efforts”, as it included the items: “I am always polite with all colleagues and teammates”, “I try to keep a good relationship with my teammates” and “I try to understand the difficulties my teammates are facing”.

The third factor was named “nutrition management efforts”, as it included the items: “I take health supplements (herbal medicine, nutritional supplements)”, “I eat food that is good for my body”, and “I pay attention to manage my diet”.

The fourth factor was named “cognitive and psychological efforts”, as it included the items: “I analyze my performance while watching videos of my games, competitions, or training”, “I analyze the performance of the best athletes while watching videos of games or competitions,” and “I engage in image training”. 

The first factor of the construct of reward was named “future stability”, as it included the six items of: “As an athlete, my future is promising”, “If I continue to perform as now, I will be recognized in the field”, “If I continue to perform, my contract period will be guaranteed”, “If I continue to perform I will be able to continue my career at the university or professional level”, “If I continue to perform, I can have a socially stable life”, “If I continue on this trajectory I will be able to reach my goals”.

The second factor was named “social support”, as it included the items of: “I have received praise and recognition from other athletes and coaches”, “I believe that I am being treated fairly based on my ability/performance”, “During training, I get attention and instruction from the instructor”, “I am fairly treated on the team”. 

Finally, the third factor was named “positive growth”, as it included the items of: “I believe I am growing as a person through the sports”, “I think it is a reward for my efforts when I get an individual award”, “I believe that improvement in my technical skills is a reward for my hard work”, “When I play without regret there are consequences for my hard work”.

## 4. Discussion 

As a social species, a balance between giving and taking matters for humans, and the concept of reward for effort is of great significance to them. For this reason, research on ERI has involved participants from various occupations. However, there has been almost no research on this topic in the field of sports, and there is no measurement scale either to evaluate the effort and reward in athletes. Therefore, based on the basic assumption that the concept of effort and reward plays an important role in athletes’ lives, the effort and reward scale for athletes was developed in this study. In this developed scale, the factors for the construct of effort included: “training reinforcement efforts”, “interpersonal relationship efforts”, “nutrition management efforts”, and “cognitive and psychological efforts”, and the factors for the construct of reward included “future stability”, “social support,” and “positive growth”. The effort scale consisted of 14 items from four factors, and the reward scale consisted of 14 items from three factors, with each item scored on a 5-point Likert scale. 

In this study, “training reinforcement efforts” appeared as a factor of effort. This result was consistent with the results of an exploratory study conducted on the efforts of Australian riders [22] reporting “physical demand” as a sub-domain of effort, as well as the results of a study employing the effort questionnaire by Siegrist et al. [33] including physical factors. The “training reinforcement efforts” factor included physical demand and the investment of time to meet such a demand. The “cognitive and psychological efforts” factor was also extracted, and included efforts invested in image training and watching training or game (competition) videos; the extraction of this factor was consistent with the results of a study on Australian riders arguing for cognitive demand in athletes [22]. In addition, the “nutrition management efforts” factor was identified for the construct of effort. Along with cognitive and psychological efforts, it reflected the efforts of athletes to make improvements based on the awareness of needs with advances in sports science. The “interpersonal relationship efforts” factor, which had not been identified in the existing ERI scale [30] or the study on Australian riders, seemed to reflect the cultural characteristics of South Korea, which are defined by a strong “we-ness”, and the tendency of Korean athletes to exhibit these cultural characteristics [34]. These four factors of “training reinforcement efforts”, “interpersonal relationship efforts”, “nutrition management efforts”, and “cognitive and psychological efforts” were also extracted by Park et al. [27] while exploring the effort perceived by athletes, without omitting any of the factors. 

“Future stability” and “social support” were extracted as the factors of reward. These results were consistent with the results of three studies: a study by Siegrist et al. [33] identifying monetary reward (wage, salary), respect (esteem), and position (promotion, security) as the factors of reward; a study [35] identifying respect, job opportunity, and job security as the factors of reward; and a study by Kathleen et al. [22] reporting position, positive feeling, victory, and new experiences as the factors of reward. Although the results were quite similar, monetary reward was not identified as a factor in this study. Park et al. [27] also extracted the benefit factor that included monetary reward as the main factor of compensation. Monetary reward may not have emerged as a factor of reward in this study’s EFA as its participants were university athletes. 

## 5. Conclusions

This study was the first attempt to develop an effort and reward scale for athletes. In the process of scale development, exploratory and confirmatory factor analysis results, consistency reliability, and test-retest reliability showed statistically significant results. Suggestions for follow-up research regarding the matters not addressed in this study include, firstly, that the discriminant validity and convergent validity of the effort and reward scale for athletes require reevaluation. This study evaluated convergent validity through the correlation of the developed scale with the self-management scale by Kim [28], which was adequate for evaluating the convergent validity for effort, but insufficient to measure the convergent validity for reward. Therefore, other questionnaires should be added to the convergent validity analysis to ensure the validity of the developed scale. In addition, while the values for the factors showed differences according to gender or athletic career, the differences were not very significant in the reward factors. This could be because the participants of this study were university athletes who had been recruited by the universities based on their performance in games (competitions) and their grades. To compensate for this limitation, the validity of the scale should be further verified by diversifying the participants. 

The effort and reward scale for athletes developed in this study is intended to be used by elite athletes at universities. Therefore, its effectiveness in extended application to younger athletes, or adult athletes who are not university athletes, needs to be evaluated. If the validity and reliability of the developed scale are ensured through the scale’s application to athletes of various ages, the scale could be employed in further research investigating related topics, such as changes in the awareness of athletes according to their efforts and rewards, and the effect of effort-reward balance on physical and psychological factors in athletes. 

## Figures and Tables

**Figure 1 ijerph-18-13396-f001:**
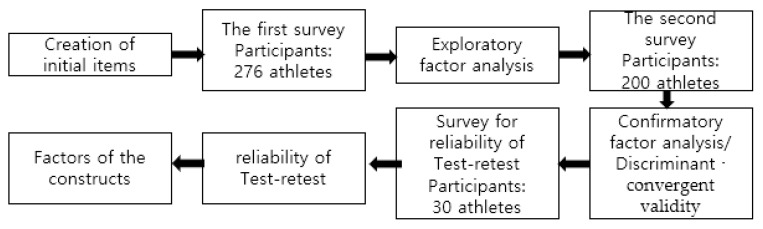
The procedure over time.

**Table 1 ijerph-18-13396-t001:** The extracted factors from effort.

Items	M(SD)	S	K	Components			
				1	2	3	4
1. I train individually (besides the team training).	3.75 (1.07)	−0.63	−0.16	0.82	−0.04	−0.01	0.09
5. I do extra-training session before and after scheduled team training.	3.50 (1.04)	−0.12	−0.53	0.90	−0.02	0.01	−0.08
6. I try more than other athletes.	3.78 (0.84)	−0.41	0.21	0.53	0.13	−0.01	0.22
11. I do additional strength training and conditioning on my own.	3.27 (0.97)	−0.05	−0.30	0.55	0.11	0.10	−0.04
21. I spend more time for training than other athletes do	3.22 (0.90)	−0.21	0.34	0.57	0.04	0.08	0.18
33. I am always polite with all colleagues and teammates.	4.16 (0.73)	−0.38	−0.66	−0.08	0.69	0.08	0.02
42. I try to keep a good relationship with the teammates.	4.30 (0.74)	−0.72	−0.27	0.01	0.84	−0.05	−0.02
45. I try to understand the difficulties my teammates are facing	4.15 (0.73)	−0.46	−0.28	0.08	0.62	0.05	0.02
17. I take some supplements (nutritional supplements).	3.46 (1.20)	−0.46	−0.66	−0.08	−0.03	0.70	0.21
37. I try to eat healthy to take care of my body.	3.92 (0.91)	−0.70	0.61	0.08	0.09	0.70	−0.12
40. I care about my nutritional management	3.29 (1.06)	−0.19	−0.38	0.23	−0.04	0.47	−0.14
16. By watching video footage, I analyze my individual performance and my competition.	3.93 (0.93)	−0.57	−0.26	0.02	0.13	0.01	0.61
19. I analyze video clips on outstanding athletes.	3.74 (0.97)	−0.64	0.17	0.11	−0.06	0.06	0.68
12. I do imagery training (visualization training).	3.95 (0.87)	−0.73	0.51	0.20	0.19	0.07	0.33
Eigenvalues				2.77	1.74	1.39	1.28
Cronbach’s α				0.85	0.77	0.67	0.66
% variance				19.8	12.41	9.91	9.16
Correlations between factors F1–F4				F1	F2	F3	F4
F1				1			
F2				0.246 ***	1		
F3				0.461 ***	0.261 ***	1	
F4				0.510 ***	0.286 ***	0.339 ***	1

S: skewness, K: kurtosis, *** *p* < 0.001.

**Table 2 ijerph-18-13396-t002:** The extracted factors from reward.

Item	M(SD)	S	K	Components		
				A	B	C
3. As an athlete, my future is promising	3.34 (0.98)	−0.26	−0.20	0.58	0.13	0.05
15. If I continue to perform as now, I will be recognized in the field.	3.48 (1.01)	−0.30	−0.41	0.64	0.06	0.07
19. If I continue to perform, my contract period will be guaranteed.	3.28 (0.95)	−0.04	0.08	0.64	0.11	0.08
26. If I continue to perform, I will be able to continue my career at the university or professional level.	3.34 (1.02)	−0.22	−0.35	0.89	−0.11	−0.01
33. If I continue to perform, I can have a socially stable life.	3.45 (0.91)	−0.22	−0.05	0.63	0.09	0.05
36. If I continue to this trajectory I will be able to reach my goals.	3.31 (1.02)	−0.11	−0.40	0.79	0.07	−0.07
1. I have received praise and recognition from other athletes and coaches.	3.52 (0.79)	0.04	0.04	0.01	0.66	0.03
2. I believe that I am being treated fairly based on my ability/performance.	3.59 (0.71)	−0.19	0.19	−0.03	0.73	−0.05
13. During training, I get attention and instruction from the instructor.	3.55 (0.86)	−0.23	0.21	−0.01	0.62	0.08
32. I am fairly treated on the team.	3.37 (0.85)	0.07	−0.13	0.13	0.67	−0.01
20. I believe that I am growing as a person through the sports.	3.95 (0.80)	−0.33	−0.26	0.12	0.06	0.50
30. I think it is a reward for my efforts when I get an individual award.	4.13 (0.79)	−0.50	−0.52	−0.03	−0.01	0.72
27. I believe that improvement in my technical skills is a reward for my hard work.	4.03 (0.77)	−0.25	−0.76	0.15	−0.01	0.67
37. When I play without regret there are consequences for my hard work.	4.06 (0.82)	−0.51	−0.22	−0.14	0.01	0.69
Eigenvalues				3.33	2.08	1.82
Cronbach’s α				0.89	0.78	0.76
% variance				23.80	14.80	13.00
Correlations between factors F1–F3				F1	F2	F3
F1				1		
F2				0.653 ***	1	
F3				0.420 ***	0.373 ***	1

S: skewness, K: kurtosis, *** *p* < 0.001.

**Table 3 ijerph-18-13396-t003:** Confirmatory analysis of effort.

	Estimate	S.E.	C.R.	SMC	SRW	CR	AVE
e 1 ← F1	1.00			0.62	0.79	0.84	0.52
e 5 ← F1	0.96	0.07	12.93	0.61	0.78		
e 6 ← F1	0.72	0.06	11.36	0.48	0.69		
e11 ← F1	0.77	0.07	10.32	0.40	0.64		
e21 ← F1	0.75	0.07	11.40	0.48	0.70		
e33 ← F2	1.00			0.42	0.65	0.75	0.50
e42 ← F2	1.14	0.13	8.61	0.58	0.76		
e45 ← F2	1.15	0.14	8.53	0.51	0.72		
e17 ← F3	1.00			0.39	0.63	0.74	0.49
e37 ← F3	0.97	0.11	8.77	0.66	0.81		
e40 ← F3	0.91	0.11	8.02	0.40	0.63		
e16 ← F4	1.00			0.51	0.71	0.72	0.46
e19 ← F4	1.04	0.12	8.48	0.47	0.68		
e12 ← F4	0.86	0.11	8.09	0.40	0.63		
Correlations	F1		F2	F3		F4	
F1	1						
F2	0.37 ***		1				
F3	0.63 ***		0.38 ***	1			
F4	0.59 ***		0.48 ***	0.46 ***		1	

*** *p* < 0.001.

**Table 4 ijerph-18-13396-t004:** Confirmatory analysis of reward.

KERRYPNX	Estimate	S.E.	C.R.	SMC	SRW	CR	AVE
r 3 ← F1	1.00			0.42	0.65	0.87	0.52
r15 ← F1	1.06	0.14	7.80	0.42	0.65		
r19 ← F1	1.09	0.12	8.90	0.55	0.74		
r26 ← F1	1.27	0.15	8.79	0.56	0.75		
r33 ← F1	1.15	0.13	8.78	0.54	0.74		
r36 ← F1	1.35	0.15	9.03	0.62	0.79		
r 1 ← F2	1.00			0.48	0.69	0.78	0.47
r13 ← F2	0.98	0.11	8.62	0.48	0.69		
r32 ← F2	1.04	0.13	8.26	0.53	0.73		
r 2 ← F2	0.73	0.10	7.60	0.39	0.62		
r20 ← F3	1.00			0.41	0.64	0.77	0.53
r27← F3	1.29	0.15	8.37	0.68	0.82		
r37 ← F3	1.21	0.16	7.50	0.44	0.66		
r30 ← F3	1.12	0.15	7.64	0.49	0.70		
Correlations	F1		F2			F3	
F1	1						
F2	0.80 ***		1				
F3	0.50 ***		0.54 ***			1	

*** *p* < 0.001.

**Table 5 ijerph-18-13396-t005:** MANOVAs of effort and reward by groups.

	Effort				Reward		
	Effort1	Effort2	Effort3	Effort4	Reward1	Reward2	Reward3
male	3.60 (0.76)	4.21 (0.60)	3.64 (0.78)	3.94 (0.70)	3.39 (0.78)	3.51 (0.63)	4.04 (0.64)
female	3.28 (0.74)	4.20 (0.62)	3.37 (0.89)	3.72 (0.72)	3.31 (0.79)	3.50 (0.63)	4.06 (0.54)
*F*	10.49 **	0.01	6.35 *	5.70 *	0.71	0.01	0.09
	*λ* = 0.93, Error *df* = 27, Hypoth *df* = 7, *F* = 2.85, *p* = 0.01						
championship	3.57 (0.80)	4.27 (0.60)	3.67 (0.81)	3.93 (0.78)	3.43 (0.75)	3.58 (0.63)	4.08 (0.61)
non-winner	3.47 (0.74)	4.16 (0.61)	3.49 (0.83)	3.83 (0.67)	3.32 (0.81)	3.46 (0.62)	4.02 (0.61)
*F*	0.26	0.17	0.07	0.28	0.28	0.12	0.42
	*λ* = 0.98, Error *df* = 27, Hypoth *df* = 7, *F* = 0.69, *p* = 0.68						
~8 years	3.49 (0.75)	4.18 (0.60)	3.45 (0.84)	3.86 (0.74)	3.37 (0.82)	3.48 (0.66)	4.11 (0.65)
9 years~	3.53 (0.78)	4.23 (0.61)	3.68 (0.79)	3.88 (0.69)	3.364 (0.74)	3.54 (0.60)	3.98 (0.56)
*F*	0.19	0.54	5.47 *	0.07	0.01	0.71	3.27
	*λ* = 0.95, Error *df* = 27, Hypoth *df* = 7, *F* = 2.02, *p* = 0.05						

* *p* < 0.05 ** *p* < 0.01.

**Table 6 ijerph-18-13396-t006:** Correlations among the sub-factors in effort, reward and self-management.

	M	SD	1	2	3	4	5	6	7
1	3.51	0.77	1						
2	4.20	0.61	0.25 ***	1					
3	3.56	0.83	0.46 ***	0.26 ***	1				
4	3.87	0.71	0.51 ***	0.29 ***	0.34 ***	1			
5	3.37	0.78	0.48 ***	0.15 *	0.43 ***	0.42 ***	1		
6	3.51	0.63	0.48 ***	0.22 ***	0.43 ***	0.37 ***	0.66 ***	1	
7	4.04	0.61	0.36 ***	0.41 ***	0.28 ***	0.33***	0.42 ***	0.37 ***	1
8	4.21	0.63	0.40 ***	0.52 ***	0.37 ***	0.41 ***	0.38 ***	0.30 ***	0.50 ***
9	4.12	0.57	0.28 ***	0.62 ***	0.31 ***	0.31 ***	0.23 ***	0.29 ***	0.49 ***
10	3.18	0.78	0.48 ***	0.16 **	0.41 ***	0.36 ***	0.45***	0.40 ***	0.18 **
11	3.92	0.59	0.57 ***	0.43 ***	0.47 ***	0.42 ***	0.44***	0.44 ***	0.43 ***
12	4.42	0.56	0.18 **	0.56 ***	0.26 ***	0.27 ***	0.13 *	0.15 *	0.49 ***
13	3.94	0.61	0.46 ***	0.44 ***	0.58 ***	0.36 ***	0.40 ***	0.38 ***	0.40 ***

* *p* < 0.05 ** *p* < 0.01 *** *p* < 0.001; 1: effort1, 2: effort2, 3: effort3, 4: effort4, 5: reward1, 6: reward2, 7: reward3, 8: mental management, 9: living management, 10: unique behavior, 11: training management, 12: interpersonal management, 13: physical care.

**Table 7 ijerph-18-13396-t007:** Test-retest of effort and reward scale.

Factors (*n* = 30)	Test		Retest		Significance Level		Correlation
	M	SD	M	SD	*t*	*p*	
effort 1	4.06	0.49	4.00	0.53	1.27	0.21	0.87 ***
effort 2	4.49	0.39	4.34	0.47	1.53	0.14	0.74 ***
effort 3	4.04	0.65	3.97	0.72	1.52	0.14	0.93 ***
effort 4	4.16	0.54	4.08	0.51	1.57	0.13	0.87 ***
reward 1	3.41	0.72	3.34	0.62	1.17	0.25	0.90 ***
reward 2	3.48	0.65	3.44	0.71	0.44	0.67	0.81 ***
reward 3	3.97	0.61	3.91	0.64	0.98	0.34	0.87 ***

*** *p* < 0.001.

## Data Availability

The data included in the present study are available upon request from corresponding author.

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
