# Peer review of "Development and Initial Validation of the Korean Effort and Reward Scale (ERS-K) for Use in Sport Contexts"

_ijerph, 2021, doi:10.3390/ijerph182413396_

Round 1

Reviewer 1 Report

First of all, I would like to congratulate the authors for validating an evaluation tool.

It should be noted that they are based on the ERI model.

In the introduction to the article, could it be explained if there is any effort and reward scale for athletes?

In the methodology section, a little more could be described: the participation of the experts, discussion rounds carried out, instruments used to assess the proposal, the type of categories and items that made up the effort and reward scale for athletes. How was the effort and reward scale for athletes scored?

They should check the style of the font on line number 336.

Remember that citations are not used in the abstract.

Why don't the results of the psychometric analysis stand out in the conclusion: Cronbach's alpha, ICC, test-retest score, etc.?

In short, an article is observed with an important contribution of statistical analysis to confirm that it is a valid and reliable tool.

Congratulations

Author Response

  1. In the introduction to the article, could it be explained if there is any effort and reward scale for athletes?

-As long as I know from previous literatures, there is no effort and reward scale for athletes. In previous studies, I decided that the existing effort-reward scale is not suitable for athletes. So the authors carried out to develop a measure of effort and reward scale for athletes.

  1. In the methodology section, a little more could be described: the participation of the experts, discussion rounds carried out, instruments used to assess the proposal, the type of categories and items that made up the effort and reward scale for athletes. How was the effort and reward scale for athletes scored?

- Thanks. The following information has been added and revised in the method section according to your suggestions.

“Universities in Seoul, Gyeonggi-do, Gangwon-do, and Jeolla-do for participation. Of the data collected from the 530 athletes, 24 datasets had the same responses throughout, had no response for more than three items, or did not include information on personal characteristics; hence, these datasets were not used in the study. Data from 276 athletes were used in the process of extracting the factors of the constructs of effort and reward (male: 192, female: 84, average age: 19.89, average athlete career: 8.51), and data from 200 athletes were used for model verification (male: 142, female: 58, average age: 20.03, average athlete career: 8.69). The remaining 30 datasets were used for the test-retest reliability analysis (male: 30, average age: 20.23, average athlete career: 9.9).”

-The content was created based on an empirical search of the efforts and rewards presented in a previous study [28]. In a study by Park et al. [28], which confirmed the concept of athlete efforts and rewards, athlete’s efforts were explored in terms of skill enhancement, self-management, cognitive psychological reinforcement, and interaction, and athlete’s rewards were extracted in terms of benefits, social support, positive outcomes, and future stability.

  1. They should check the style of the font on line number 336.

-I have done

  1. Remember that citations are not used in the abstract.

-Yes. I have deleted them in the abstract.

  1. Why don't the results of the psychometric analysis stand out in the conclusion: Cronbach's alpha, ICC, test-retest score, etc.

- I added the following sentence in Conclusion. “In the process of scale development, exploratory and confirmatory factor analysis results, consistency reliability, and test-retest reliability showed statistically significant results.”

Reviewer 2 Report

Generally an interesting paper that is can be an important contribution.  However, there are a number of points of clarification that would improve the paper.

Methods

The measure development process is unclearly elucidated.  It sounds like the research team elicited feedback from experts.  What was their expertise in?  How many people provided feedback?  What was the nature of the feedback elicitation process and the feedback received.
In addition, how was feedback was elicited from research participants is not clear as well.  Were there 40 people who engaged in some sort of process of responding to the survey items?

You describe an EFA and CFA.  Can you also characterize the samples that you used to create this analysis (e.g. timeline the data was collected, differences or similarities in demographics, strategies for used for recruitment).  I know you briefly describe the sample above but additional clarity will help readers understand your results more completely.  In addition, it would be helpful for you to describe more about the EFA.  How many factors were in the model, what were the correlations between the items?

For your reliability evaluation, what was your sample size?

You mention IRB approval for the survey section, was the rest of the study not IRB approved?

Further, it would be helpful to provide a correlation table of the items in your final scale as part of the EFA.  This would provide researchers with a sense of how the entire scale fits together that helps in interpreting the rest of the findings.

Further, you choose cutoffs (which are reasonable) but you should provide citations for these.

Discussion seems appropriate.

Author Response

  1. The measure development process is unclearly elucidated. It sounds like the research team elicited feedback from experts. What was their expertise in? How many people provided feedback? What was the nature of the feedback elicitation process and the feedback received.

- Expert opinions were collected at a meeting of eight experts including two professors and six doctors in sports and Exercise psychology for creating the initial items of the questionnaire and deciding the factor names after searching for the contents.

  1. In addition, how was feedback was elicited from research participants is not clear as well. Were there 40 people who engaged in some sort of process of responding to the survey items?

- I expressed it in a picture so that you can clearly check it.

  1. You describe an EFA and CFA. Can you also characterize the samples that you used to create this analysis (e.g. timeline the data was collected, differences or similarities in demographics, strategies for used for recruitment). I know you briefly describe the sample above but additional clarity will help readers understand your results more completely. In addition, it would be helpful for you to describe more about the EFA. How many factors were in the model, what were the correlations between the items?

-I added the following sentences. “The content was created based on an empirical search of the efforts and rewards presented in a previous study [28]. In a study by Park et al. [28], which confirmed the concept of athlete efforts and rewards, athlete’s efforts were explored in terms of skill enhancement, self-management, cognitive psychological reinforcement, and interaction, and athlete’s rewards were extracted in terms of benefits, social support, positive outcomes, and future stability.”

  1. For your reliability evaluation, what was your sample size?

- The number of research participants used in the reliability analysis conducted in the exploratory factor analysis was 276, and the number of research participants used in the actuality analysis conducted in the confirmatory factor analysis was 200. And 30 athletes participated in the survey for test-retest reliability.

  1. You mention IRB approval for the survey section, was the rest of the study not IRB approved?

I redescribed the IRB approval process. “This research process was conducted with the approval of the Institutional Review Board (IRB) of Seoul National University of Science and Technology (IRB approval number: 2020-0003-01).”

  1. Further, it would be helpful to provide a correlation table of the items in your final scale as part of the EFA. This would provide researchers with a sense of how the entire scale fits together that helps in interpreting the rest of the findings.

- It is presented in the text.

  1. Further, you choose cutoffs (which are reasonable) but you should provide citations for these.

- It is presented in the text.

Round 2

Reviewer 2 Report

This paper is significantly improved.  Thank you for addressing my concerns.